# Impact of Donor Human Milk in the Preterm Very Low Birth Weight Gut Transcriptome Profile by Use of Exfoliated Intestinal Cells

**DOI:** 10.3390/nu11112677

**Published:** 2019-11-05

**Authors:** Anna Parra-Llorca, María Gormaz, Sheila Lorente-Pozo, Maria Cernada, Ana García-Robles, Isabel Torres-Cuevas, Julia Kuligowski, Maria Carmen Collado, Eva Serna, Máximo Vento

**Affiliations:** 1Neonatal Research Group, Health Research Institute La Fe, 46026 Valencia, Spain; annaparrallorca@gmail.com (A.P.-L.); gormaz_mar@gva.es (M.G.); slopo@alumni.uv.es (S.L.-P.); mariacernada@gmail.com (M.C.); garcia.anarob@gmail.com (A.G.-R.); Maria.I.Torres@uv.es (I.T.-C.); julia.kuligowski@uv.es (J.K.); 2Division of Neonatology, University and Polytechnic Hospital La Fe, 46026 Valencia, Spain; 3Department of Biotechnology, Institute of Agrochemistry and Food Technology, Spanish National Research Council (IATA-CSIC), 46980 Valencia, Spain; mcolam@iata.csic.es; 4Department of Physiology, Faculty of Medicine, University of Valencia, 46010 Valencia, Spain

**Keywords:** prematurity, mother’s milk, donor milk, genetics, intestinal cells, inflammation, oxidative stress

## Abstract

Background: Own mother’s milk (OMM) is the optimal nutrition for preterm infants. However, pasteurized donor human milk (DHM) is a valid alternative. We explored the differences of the transcriptome in exfoliated epithelial intestinal cells (EEIC) of preterm infants receiving full feed with OMM or DHM. Methods: The prospective observational study included preterm infants ≤ 32 weeks’ gestation and/or ≤1500 g birthweight. Total RNA from EEIC were processed for genome-wide expression analysis. Results: Principal component analysis and unsupervised hierarchical clustering analysis revealed two clustered groups corresponding to the OMM and DHM groups that showed differences in the gene expression profile in 1629 transcripts. The OMM group overexpressed lactalbumin alpha gene (*LALBA*), Cytochrome C oxidase subunit I gene (*COX1*) and caseins kappa gene (*CSN3*), beta gene (*CSN2*) and alpha gene (*CSN1S1*) and underexpressed Neutrophil Cytosolic Factor 1 gene (*NCF1*) compared to the DHM group. Conclusions: The transcriptomic analysis of EEIC showed that OMM induced a differential expression of specific genes that may contribute to a more efficient response to a pro-oxidant challenge early in the postnatal period when preterm infants are at a higher risk of oxidative stress. The use of OMM should be strongly promoted in preterm infants.

## 1. Introduction

Prematurity causes annually about 1 million neonatal deaths worldwide and is the second cause of neonatal and under five-years childhood mortality [1,2]. Preterm infants have an immature antioxidant defense system [3], and clinical therapies, such as radiation, parenteral nutrition, mechanical ventilation, or antibiotics increase oxidative stress-related conditions [4,5]. Moreover, preterm infants’ stabilization in the delivery room frequently requires the provision of positive pressure ventilation and oxygen supplementation, both leading to the generation of reactive oxygen species (ROS) and oxidative stress [6]. Breast milk provides antioxidant protection [7], and therefore, premature infants fed human milk (HM) had lower levels of oxidative stress-related biomarkers than those fed with formula [8]. Donor human milk (DHM) is a valid alternative when their own mother’s milk (OMM) is unavailable. DHM protects against necrotizing enterocolitis (NEC) [9]; however, pasteurization and freezing processes alter the biological activity and concentration of several biological factors and biomolecules present in DHM [10]. 

In a previous study, we analyzed the transcriptome profile from exfoliated epithelial intestinal cells (EEIC) in preterm neonates with sepsis, and results showed that infection contributed to changes in the gene expression favoring a pro-oxidant status in the intestinal lumen [11]. To our knowledge, there are no studies showing the differences in fresh preterm own mother’s milk compared to pasteurized human milk on the transcriptome of preterm infants’ intestine. Accordingly, we studied the transcriptomic profile in EEIC collected from preterm infants fed OMM and compared to those fed with pasteurized DHM.

## 2. Material and Methods

### 2.1. Study Design and Patients’ Characteristics

This is a prospective, observational, cohort study performed in the Division of Neonatology of the University and Polytechnic Hospital La Fe (HUiP) from January 2016 to December 2017. The Neonatal Division protocol strongly supports breastfeeding to preterm infants born ≤32 weeks or ≤1500 g birth weight. OMM is the first choice, and DHM is used as a supplement when there is not enough OMM available or refusal of parents to breastfeed. 

The study was approved by the Scientific and Ethics Committee for Biomedical Research (CEIm) (#2015/0371) of the HUiP. Parents signed an informed consent form.

Demographic, perinatal, clinical, and analytical data were recorded and matched according to the type of feeding (see Table 1). Enrolled patients had never been fed Formula Milk. Full OMM feeding was attempted in all cases, but not achieved in all patients. Two groups were recruited according to the main feeding type accounting for ≥80% of the nutritional intake by volume with either OMM or DHM. When necessary, DHM or OMM was provided respectively to complete the resting volume.

Preterm infants were fed upon admission to the NICU small volumes of trophic enteral nutrition (between 15–30 mL/kg/day), which was progressively increased until full enteral nutrition was reached at day 10^th^–12^th^ after birth. Nutrition was complemented with intravenous parenteral nutrition, which was gradually decreased until it was withdrawn. Fecal samples were collected when preterm infants achieved complete enteral nutrition (150 mL/kg/day). A stool sample per child was collected.

Daily intake was provided with OMM or DHM to achieve the prescribed feeding volume. Patients of the DHM group only received milk from one donor. The nutritional intake was monitored, but never influenced by this observational study. Exclusion criteria included mixed breastfeeding, antibiotic therapy, chromosomopathies, major congenital malformations or abdominal surgery. DHM was pasteurized following the Holder method (62.5 °C degrees for 30 min).

Fecal samples were directly collected (minimum 5 g) from the diaper when full enteral feeding was achieved. Samples were frozen and stored at −80 °C until analysis.

RNA extraction and transcriptomic analysis of samples from EEIC.

Total RNA was isolated directly from fecal samples using the Trizol reagent (TRIzol, Invitrogen, Thermo-Fisher Scientific, Carlsbad, CA, USA), followed by a polyA+ RNA enrichment step in order to eliminate contaminating DNA and bacterial RNA using the mRNA-ONLY™ Eukaryotic mRNA Isolation Kit (Epicentre Analytics Inc., Etobicoke, ON, Canada) according to the manufacturers´ instructions. RNA integrity number (RIN) was tested employing the 2100 Bioanalyzer (Agilent Technologies, Santa Clara, CA, USA), and RNA concentration and purity were determined using a spectrophotometer (GeneQuant, GE Healthcare Biosciences, German). The ratio 260/280 and the integrity of RNA between both populations were compared using T-Student test. RIN values >6 were considered optimal for hybridization and were statistically similar (*p*-value = 0.6).

The synthesis of cDNA and cRNA, labeling, hybridization, and scanning of the samples were performed according to the WT Plus Reagent Kit Manual (Thermo Fisher Scientific, Madrid, Spain). To prepare the hybridization cocktail, 5.5 μg of fragmented biotinylated cRNA was used. Subsequently, this cocktail was hybridized onto the Clariom S Human microarray for 16 h at 45 °C (Thermo Fisher Scientific, Madrid, Spain). This microarray contains more than 20,000 well-annotated genes. The microarray used is specific only for the human organism.

The raw data were imported into Partek Genomics Suite v6.6 (Partek, Inc., St. Louis, MO, USA) as CEL files. Raw data were pre-processed, which included background correction, normalization, and summarization using robust multi-array average (RMA) analysis and then log2-transformed. 

### 2.2. Real-Time RT-PCR Validation

The Cytochrome C oxidase subunit 1 (*COX1*) (Hs02596864_g1) and Neutrophil Cytosolic Factor 1 (*NCF1*) (Hs00165362_mL) significant differentially expressed transcripts were validated by using a single real-time reverse transcription polymerase chain reaction (RT-PCR) using TaqMan Gene Expression Assays probes (Applied Biosystems, Foster City, CA, USA), according to the manufacturer’s instructions. The endogenous reference gene used was glyceraldehyde-3-phosphate dehydrogenase (housekeeping control Hs02786624_g1). RNA samples were reverse transcribed using random hexamers and MultiScribe reverse transcriptase (Applied Biosystems). After complementary DNA synthesis, real-time polymerase chain reaction (RT-PCR) was carried out using the QuantStudio 5 Real-Time PCR System (Applied Biosystems). The same samples used in microarray analysis were run in triplicate, and fold-changes were generated for each sample by calculating 2^−ΔΔCT^ [12].

Statistical significance of the differences was evaluated by the Mann-Whitney test. The level of significance was set at *p* < 0.05.

### 2.3. Statistical Analysis

Categorical variables were compared using χ^2^ or Fisher’s exact test (two-tailed). Continuous variables were expressed as mean ± SD or medians with the interquartile range, depending on data distribution. Two-tailed Student’s *t* or Mann-Whitney *U* test and analysis of variance (ANOVA) or Kruskal-Wallis were used to compare two or >two groups as appropriate. Kolmogorov-Smirnov analysis was performed to test the normal distribution of the data. Data analysis was performed using SPSS version 20.0 (IBM, Madrid, Spain). Significance was considered for a *p*-value <0.05. 

Unsupervised analysis of data was carried out employing principal component analysis (PCA) and unsupervised hierarchical cluster analysis. 1-way ANOVA (*p*-value < 0.05) was applied to identify differentially expressed genes between OMM vs. DHM groups. Finally, the selected differentially expressed genes were imported into Pathway Studio version 10 (Elsevier Inc, Rockville, MD, USA) to identify relevant biological processes.

## 3. Results

### 3.1. Patients’ Characteristics

A total of 47 preterm infants ≤32 weeks’ gestation and/or ≤1500 g birth weight receiving OMM (*N* = 27) or DHM (*N* = 20) were included in the study (Figure 1). No differences in prenatal demographic characteristics or confounders during the hospitalization were found between both groups (Table 1).

### 3.2. PCA of Gene Expression from EEIC of Preterm Infants

The PCA scores plot allowed to observe the distribution of fecal samples from newborns receiving OMM or DHM according to their transcriptome (see Figure 2). 

Our data were analyzed by PCA to determine the significant sources of variability of high-dimensional data, and thus, simplifying the identification of patterns and sources of variability in an extensive data set in a tridimensional fashion. The PCA scores plot depicted in Figure 2 shows a high similarity between the OMM and DHM groups; however, two different clustered groups that correspond to the OMM and DHM groups can be clearly appreciated. The variability between groups was 12.1%.

### 3.3. EEIC Transcriptome Analysis in the OMM Versus DHM Groups 

Differentially expressed genes (DEG) derived from ANOVA (*p*-value < 0.05) analysis revealed statistically significant changes in 1629 transcripts of EEIC in neonates fed OMM vs. DHM. Out of these, 807 genes were up-regulated (49.5%), and 822 were down-regulated (50.5%) in the OMM groups, when compared with the DHM group (see Appendix A, Volcano plot). Unsupervised hierarchical clustering analysis confirmed the differential intensity of gene expression between both groups (see Figure 3). Figure 3 represents in blue and red, the statistically significant underexpressed and overexpressed genes, respectively. Preterm infants fed OMM are clustered in the lower zone and fed DHM are clustered in the upper zone. No outliers are present despite being an unsupervised hierarchical clustering. The prevalence of yellow reveals that fold-change changes are subtle reconfirming PCA findings.

### 3.4. Biological Analysis

According to Pathway Studio v10, most of the changes in the gene expression between the two groups were related to transcription (*p*-value = 3.62 × 10^−24^). In the Appendix A shows in detail the top 10 biological processes filtered for enrichment *p*-value, and Appendix A the expression of the 1629 differentially expressed genes.

### 3.5. Biomarkers in Preterm Infants According to Type of Nutrition

The candidate genes to explain the transcriptomic changes between OMM versus DHM are represented in Table 2. The criterion for choosing the most over-and-underexpressed genes in premature neonates fed with OMM compared with those fed with DHM, was the absolute value of fold-change of *|1.6|*.

Gene expression of Lactalbumin alpha (*LALBA*) and caseins kappa (*CSN3*), beta (*CSN2*), and alpha (*CSN1S1*) were overexpressed in preterms fed with OMM vs. DHM. Cytochrome C oxidase subunit I (*COX1*) was also one of the genes with a higher fold-change in the OMM vs. DHM. Neutrophil Cytosolic Factor 1 (*NCF1*) was the most underexpressed gene in the OMM group, thus, explaining the lack of activation of inflammatory pathways, non-activating cytokines formation, and blocking of ROS generation. Figure 4 illustrates changes in gene expression profile found in EEI cells of preterm infants fed OMM vs. DHM. 

*COX1:* Cytochrome C oxidase subunit I, *HSPA8:* Heat Shock Protein Family A (Hsp70) Member 8, *PPARA:* Peroxisome Proliferator Activated Receptor Alpha

*NFkB:* Nuclear Factor kappa Beta, *NADH dehydrogenase*: Reduced nicotinamide adenine dinucleotide dehydrogenase. *SIRT6:* Sirtuin 6. *NCF1:* Neutrophil Cytosolic Factor 1.

We performed real-time RT-PCR from RNA total of fecal samples from OMM and DHM to validate the biomarker genes *COX1* and *NCF1* (Figure 5).

The real-time RT-PCR data confirmed significantly higher mRNA levels of *COX1* in OMM with respect DHM. In line with microarray data, this validation also showed statistically significant underexpression of *NCF1* in OMM compared with DHM (shown in Appendix A). 

## 4. Discussion

Chapkin et al. [13], for the first time, employed stool samples containing intact epithelial cells to quantify intestinal gene expression profiles in infants that exclusively received breast milk or formula at three months of age. Donovan et al. studied exfoliated intestinal cells within a term and also preterm population newborn babies fed human milk versus formula and found a differential expression of ∼1200 genes, including genes regulating intestinal proliferation, differentiation, and barrier function. Of note, when term and preterm infants were compared, pathways associated with immune cell function and inflammation were up-regulated in preterm, whereas cell growth-related genes were up-regulated in the term infants [14]. Moreover, Knight et al. [15] applied next-generation sequencing to analyze differences in gene expression between term and preterm infants. However, to the best of our knowledge, this is the first report comparing the gut transcriptome in preterm babies fed OMM or DHM. For this purpose, we employed a previously validated, non-invasive sample collection method [11]. According to the World Health Organization (WHO), breastfeeding fulfills the requirements for infants’ growth and development optimally [16]. In preterm infants, OMM has been associated with improved growth and cognitive development, reduced risk of NEC, and late-onset sepsis [17]. However, mothers who deliver preterm infants are often unable to breastfeed successfully [18]. When OMM is unavailable, or insufficient DHM has become the preferred alternative for the nutrition of preterm infants, because it confers clinical advantages over formula-feeding, such as reducing the incidence of NEC [19,20]. 

In the present study, the feeding of preterm infants with either OMM or DHM caused significant differences, as shown in the PCA scores plot. In fact, the transcriptional profile of around 1600 genes in a panel over 20,000 genes differed between both groups. The candidate genes in which the whole gene expression analysis of EEI cells showed significant changes were alpha-lactalbumin (*LALBA*), caseins kappa (*CSN3*), beta (*CSN2*), and alpha (*CSN1S1*), cytochrome c oxidase subunit I (*COX1*), and Neutrophil Cytosolic Factor 1 (*NCF1*).

DHM is collected, processed, and stored in milk banks. Milk bank guidelines recommend Holder pasteurization (62.5 °C degrees for 30 min) to inactivate viral and bacterial agents [10]. Pasteurization does not substantially alter the pattern of the gut microbial colonization. Hence, in a recent study, it was shown that preterm infants fed DHM or OMM had a similar pattern of gut microbiome, as opposed to formula feeding [21]. Differences in the intestinal microbiota induced by a different diet [20], could be a potential mediating factor to induce the changes in the profile of gene expression. Pasteurization has unwanted secondary effects upon human milk, such as killing cells and bacteria, and modifying the structure and function of specific proteins, lactoferrin, lysozyme, cytokines, and vitamins. However, other components with biological relevance, such as oligosaccharides and polyunsaturated and long-chain fatty acids (LCPUFA), are preserved [22,23]. Pasteurization affects nutritional, microbiological, metabolic and immunological properties, and therefore, the clinical benefits are still a matter of debate [22,23,24,25,26,27].

Proteins act as biologically active molecules to assist in nutrient digestion and absorption, confer protection against pathogens, and modulate the immune maturation and immune response [28,29,30]. The HM proteome is complex and includes 2500 different identified proteins [31]. Alfa-lactalbumin, lactoferrin, serum albumin, and caseins are amongst the most abundant proteins and account for approximately 85% of the total protein content of HM [32]. Our results revealed overexpression of different types of caseins and lactalbumin genes in preterm infants fed with OMM. These changes are consistent with the higher protein content of milk obtained from preterm mothers compared to mothers who deliver at term and donate mature milk to the bank [33].

On the other side, the immune system in early life undergoes intense and rapid changes [34]. Similar patterns of distinct Toll-like-receptor-mediated immune responses come to light when innate immune development at the beginning of life is compared with that towards the end of life [35]. Reactive oxygen species (ROS) can be beneficial because they are used by the immune cells, especially monocytes and macrophages, to kill pathogens [36]. Oxidative stress is defined as an imbalance between the production of free radicals (FR) and the reduction capacity of the antioxidant defense system of the organism in favor of the former [3]. Oxidative stress triggers the expression of proinflammatory genes through the activation of the transcription factor NFkB [3,37,38]. ROS, cytokines and specific genes, such as NCF1, promote the translocation of NFκB to the nucleus where it binds to specific target genes that play a central role in inflammatory [39] and immune responses [40], and cytokine secretion by numerous cell types [41,42,43,44,45,46]. As shown in Figure 4, in preterm infants fed OMM NCF1 expression is down-regulated and concomitantly COX1 up-regulated. These changes may explain why fresh HM has higher anti-inflammatory and antioxidant properties than pasteurized milk [47]. The protein encoded by this NCF1 is a cytosolic subunit of neutrophil NADPH oxidase. This oxidase is a multicomponent enzyme that is activated to produce superoxide anion, and subsequently, oxidative stress [48]. Down-regulation of NCF1 reduces cytokine production, thus, playing an essential role in modulating autoimmunity, immunologic processes, inflammation, proliferative responses and apoptosis [49,50,51,52]. As shown in Figure 4, premature infants fed OMM overexpress COX1 compared with those fed with DHM. COX1 regulates the electron flow across the mitochondrial electron transport chain and the formation of H_2_O, thus, reducing free radical generation, and subsequently, the activation of pro-inflammatory pathways. [49]. Moreover, as also shown also in Figure 4, COX1, through the interaction with the chaperone protein HSPA8, also plays a vital important role under pro-oxidant conditions, such as hypoxia-reoxygenation in the inactivation of NFκB and pro-apoptotic pathways [50,51]. In addition, peroxisome proliferator-activated receptor alpha (PPAR-α) also inhibits NFκB, but not stress-activated protein kinase [52,53].

The transcriptomic analysis in EEI cells of preterm infant’s intestine receiving two different types of HM, preterm non-pasteurized and term pasteurized, has allowed the visualization genome expression changes. These results were obtained in fecal samples, and therefore, reflected local changes located in the gut of preterm infants. We may speculate that metabolites that are derived from the digestion of OMM or DHM exert a specific stimulus upon exfoliated epithelial intestinal cells eliciting a differentiated genetic response.

We acknowledge the sample size as a limiting factor; however, our NICU promotion of the use of OMM explains the difficulties for enrolling patients fed almost exclusively (≥80%) with DHM. Despite this limitation, our study shows differences in genes involved in oxidative stress and inflammation in the transcriptome profiling of EEIC of preterm infants fed with OMM as compared to DHM. The use of non-invasive techniques in this vulnerable population should also be underpinned. Finally, our study stresses the need for optimizing pasteurization methods that enhance the preservation of essential components of HM, and consequently, improve the quality of DHM.

## 5. Conclusions

The transcriptomic analysis performed in EEIC of preterm infants has been used to visualize expression changes in the gut induced by two different types of human milk, fresh own preterm mother’s milk, and donated term and pasteurized human milk. Our results, although preliminary, suggest that preterm infants fed OMM could have a better response against oxidative stress and inflammation, due to the differential expression of highly specific genes, such as *COX1*, *PPAR-alpha*, *NCF1* and *NFκB*. The use of OMM should be promoted, and improvement in pasteurization techniques should be implemented.

## Figures and Tables

**Figure 1 nutrients-11-02677-f001:**
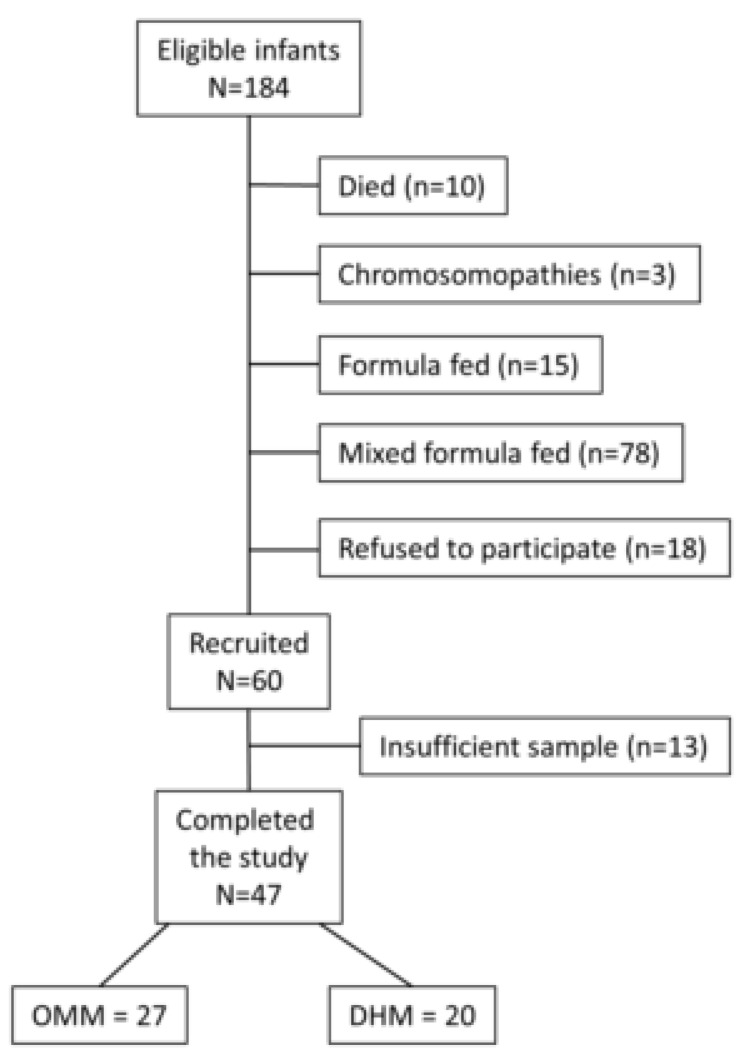
Flowchart of patients’ recruitment. A total of 47 preterm infants <32 weeks’ gestation or <1500 g were recruited. Out of these, 27 were fed with their own mother’s milk (OMM) and 20 with pasteurized donated human milk (DHM).

**Figure 2 nutrients-11-02677-f002:**
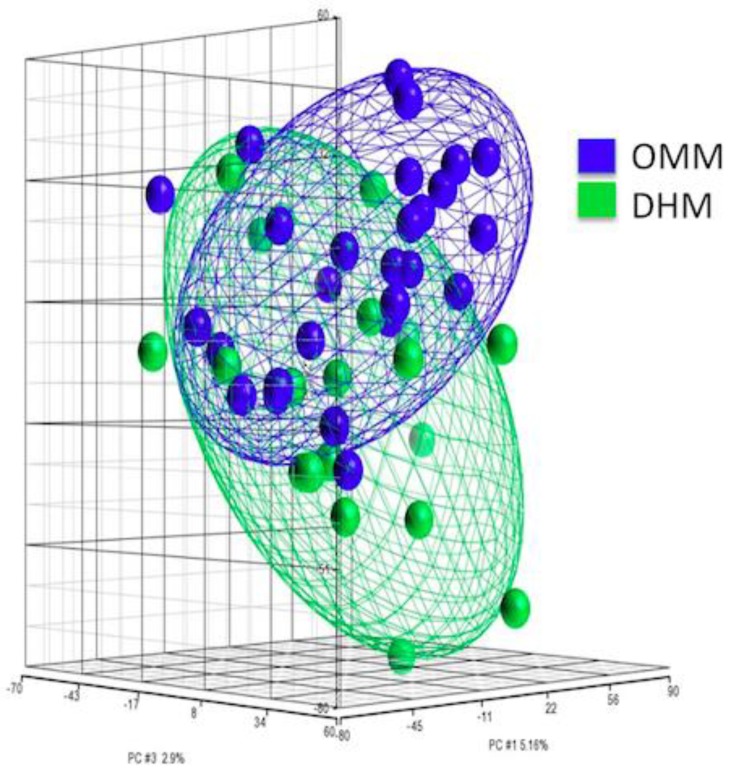
Principle components analysis (PCA) displays clear spatial separation of variations in expression values in the two groups of samples identified by unsupervised hierarchical clustering. In the 3-dimensional plot, the three principal components PC#1, #2 and #3 of all samples with over 20,000 well-annotated genes and their respective variations are expressed on the x-, y- and z-axis. The total percentage of PCA mapping variability is 12.1%. Each data point represents one sample. The ellipsoids highlight the portioning of the different samples. Assignment of samples by color: OMM (blue) and DHM (green).

**Figure 3 nutrients-11-02677-f003:**
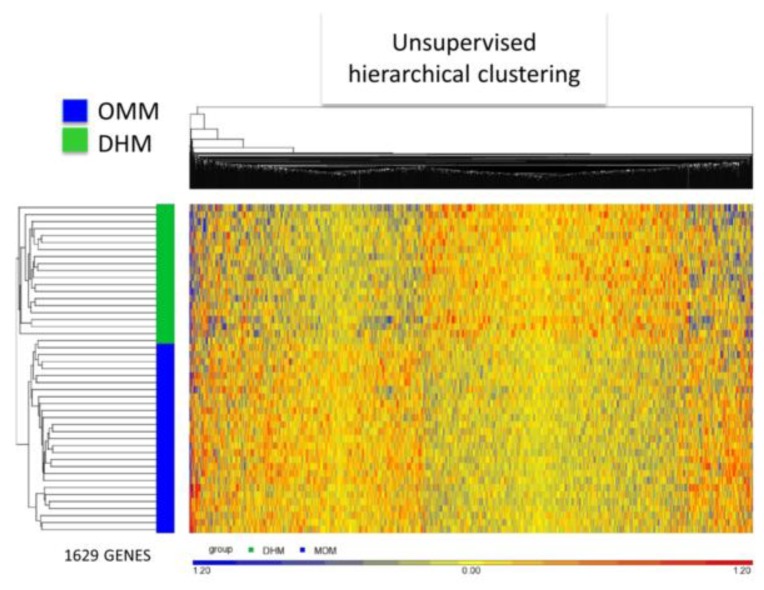
Heatmap of 27 preterm infants fed their own mother’s milk (OMM; blue bar) and 20 preterm infants fed donated human milk (DHM; green bar) and 1629 genes derived from ANOVA. Each line represents a patient and each column a gene. Overexpressed genes are represented in red and underexpressed genes in blue.

**Figure 4 nutrients-11-02677-f004:**
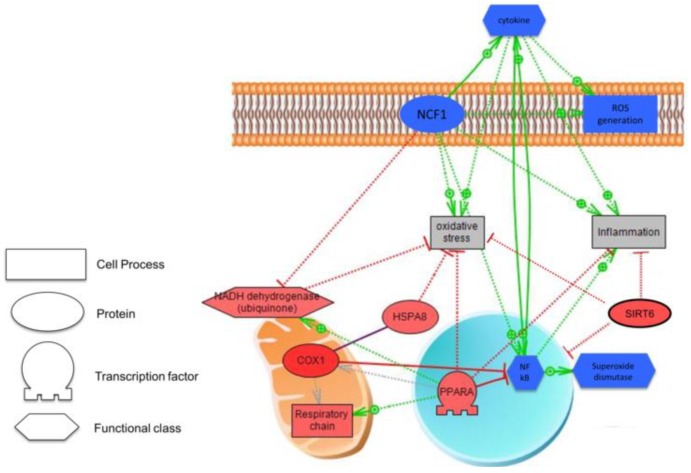
Model describing changes in gene expression profile observed in exfoliated epithelial intestinal cells (EEIC). Up-regulated (in red) and down-regulated (in blue) genes in preterm infants fed OMM vs. DHM. The solid line means expression and dashed line means regulation. Relations are colored by effect: Red expresses a negative effect, and green represents a positive effect. Grey means we can observe the unknown effect.

**Figure 5 nutrients-11-02677-f005:**
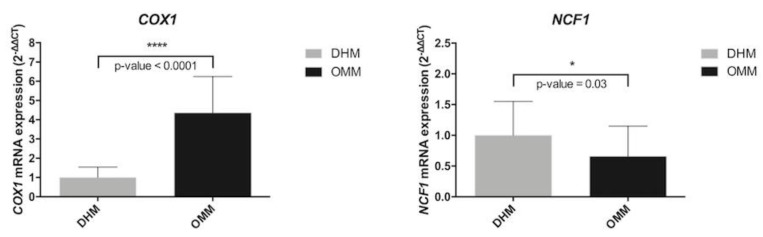
Real-time RT-PCR from RNA total of fecal samples from OMM and DHM to validate the biomarker genes *COX1* and *NCF1.*

**Table 1 nutrients-11-02677-t001:** Demographic, perinatal, clinical, and analytical data of preterm infants < 32 weeks’ gestation receiving own mother’s milk (OMM) or pasteurized donor human milk (DHM).

	OMM (*N* = 27)	DHM (*N* = 20)	*p*-Value
GA weeks, (median; 5–95% CI)	29 (28–30)	30 (29–31)	0.07
Antenatal Steroids full course, n (%)	25 (92.6)	20 (100)	0.21
Sex male, n (%)	21 (77.7)	11 (55)	0.1
Type of delivery, n (%)			
- Vaginal	13 (48.1)	10 (50)	0.9
- C- Section	14 (51.8)	10 (50)	
Birth weight (g), mean (SD)	1206 (264)	1315 (193)	0.22
Apgar 1 min (median; 5–95% CI)	7 (6–9)	7 (6–9)	0.48
Apgar 5 min (median; 5–95% CI)	9 (8–19)	9 (8–9)	0.28
Age (d) at sample collection, (median; 5–95% CI)	9 (7–11)	8 (6–10)	0.52
Chorioamnionitis, n (%)	4 (14.8)	2 (10)	0.62
Persistent ductus arteriosus, n (%)	8 (29.6)	6 (30)	0.98
Antibiotic therapy, n (%)	11 (40.7)	6 (30)	0.45
Bronchopulmonary Dysplasia, n (%)	1 (3.7)	0 (0)	0.38
Intraventricular hemorrhage, n (%)	6 (22.2)	3 (15)	0.53
Necrotizing enterocolitis, n (%)	1 (3.7)	0 (0)	0.38
Retinopathy of prematurity, n (%)	1 (3.7)	0 (0)	0.38

**Table 2 nutrients-11-02677-t002:** The candidate genes with the highest fold-change between preterm infants < 32 weeks’ gestation/<1.500 g fed their own mother’s milk versus donated human milk in the neonatal period.

Gene Symbol	Gene Name	Fold-Change	*p*-Value
*LALBA*	Lactalbumin alpha	2.9	0.002
*CSN3*	Casein kappa	2.6	0.002
*CSN2*	Casein beta	2.1	0.009
*COX1*	Cytochrome C oxidase subunit1	2.1	0.030
*CSN1S1*	Casein alpha-s1	1.7	0.008
*NCF1*	Neutrophil cytosolic factor 1	−1.6	0.040

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
