# Peer review of "Impact of Donor Human Milk in the Preterm Very Low Birth Weight Gut Transcriptome Profile by Use of Exfoliated Intestinal Cells"

_nutrients, 2019, doi:10.3390/nu11112677_

Round 1

Reviewer 1 Report

This is a very interesting and informative study and is well written.

I have just a few questions/clarifications:

Page 5, second pp, last sentence: I am assuming that the 80% figure for percent of intake means that from birth up to the time of study sampling (when reaching 150mL/kg/day of HM feedings) HM accounted for 80% of the nutritional intake by volume. Is this correct, please clarify.

Pages 11-12, with regard to figure 3: on page 11 it states that the upper zone includes data for the OMM group (indicated by the green bar) while the lower zone includes data for the DHM group (indicated by the blue bar) but the key in the upper left of the figure reverses those colors. In addition the green bar appears to include data for 20 subjects while the blue bar includes data for 27 subjects.

Page 13, Table 2, column 3: not being familiar with the notation for the fold change it is unclear to me what the significance of a positive versus a negative  number in this column means. In the pp below the figure it states that LALBA, CSN3, CSN2, and NCFI were all underexpressed by pasteurization while COX1 had a high fold change but only NCF1 has a negative number in column 3 while all others are positive and LALBA has the highest positive number. Can you clarify?

Page 14, Figure 4: some of the abbreviations in the figure are not defined. 

Author Response

Thank you in advance.

Reviewer 2 Report

Nutrition support of the preterm infant is critical for growth, development, and future health and well-being.  Yet, this is an area of research in need of much more attention.  The authors explore an area of importance, understanding how the intestine, and particularly the epithelial cells, respond to source of nutrition.  This study is a logical extension of the investigators’ previous research.  The following comments are provided to help the authors understand some aspects that may need additional consideration.

The introduction is well written

Methods:

a.What is meant by “small volumes”? Can an amount be provided (ml/kg)? b.The term “supplemented with OMM of DHM” infers something else was also fed.

c.BM contains living cells. Have the authors evaluated the transcriptome of fresh vs pasteurized BM?

Results are well presented and clear

Discussion

a.How do the present results correspond with those of the Donovan and Knight groups?

b.The finding for OMM increasing expression of caseins and lactalbumin is puzzling and needs to be explained. This implies the intestinal epithelium expresses these genes.  Is it possible the OMM includes exfoliated mammary gland cells that do express these genes?

c.Is there a way to determine from which region of the bowel the exfoliated cells originate? Small or large bowel?  Exfoliated cells from each region are likely to have specific patterns of gene expression.

d.Page 16: the authors make a comment that infers because BM was used for both groups, the microbiome would not be a factor. This possibility can’t be ruled out.  OMM would be much more like colostrum than the mature DHM and this can influence the early and rapidly developing microbiome.

e.Minor consideration: spell out cytochrome oxidase rather than use COX1 (which is not in the abbreviations list). This reviewer immediately thought of ‘cyclooxygenase 1’.  Avoid abbreviations that could elicit such misinterpretations.    

Author Response

Thank you in advance.
